# Nrf2 signaling promotes cancer stemness, migration, and expression of ABC transporter genes in sorafenib-resistant hepatocellular carcinoma cells

**Luping Gao, Yuji Morine\*, Shinichiro Yamada, Yu Saito, Tetsuya Ikemoto, Kazunori Tokuda, Chie Takasu, Katsuki Miyazaki, Mitsuo Shimada**

Department of Surgery, Institute of Biomedical Sciences, Tokushima University Graduate School, Tokushima, Japan

\* ymorine@tokushima-u.ac.jp

## Abstract

**Data Availability Statement:** All relevant data are within the manuscript and its Supporting information files.

### Background and aim

As a multiple tyrosine kinase inhibitor, sorafenib is widely used to treat hepatocellular carcinoma (HCC), but patients frequently face resistance problems. Because the mechanism controlling sorafenib-resistance is not well understood, this study focused on the connection between tumor characteristics and the Nrf2 signaling pathway in a sorafenib-resistant HCC cell line.

### Methods

A sorafenib-resistant HCC cell line (Huh7) was developed by increasing the dose of sorafenib in the culture medium until the target concentration was reached. Cell morphology, migration/invasion rates, and expression of stemness-related and ATP-binding cassette (ABC) transporter genes were compared between sorafenib-resistant Huh7 cells and parental Huh7 cells. Next, a small interfering RNA was used to knock down Nrf2 expression in sorafenib-resistant Huh7 cells, after which cell viability, stemness, migration, and ABC transporter gene expression were examined again.

### Results

Proliferation, migration, and invasion rates of sorafenib-resistant Huh7 cells were significantly increased relative to the parental cells with or without sorafenib added to the medium. The expression levels of stemness markers and ABC transporter genes were up-regulated in sorafenib-resistant cells. After Nrf2 was knocked down in sorafenib-resistant cells, cell migration and invasion rates were reduced, and expression levels of stemness markers and ABC transporter genes were reduced.

**Funding:** This study was partly supported by Research Program on Hepatitis from the Japan Agency for Medical Research and Development (AMED) Grant Numbers JP19fk0210048 and JP20fk0210048. And co-authors (Shimada M) received a research grants from Grant-in-Aid for Scientific Research (B) (KAKENHI), Project code: 18K02871, during the conduct of the study.

**Competing interests:** We state the potential conflicts of interest with regard to our study as follows: Mitsuo Shimada declares receiving unrestricted research grant from Bayer Yakuhin, CO. LTD., Japan. All other authors report no conflict of interest. This does not alter our adherence to PLOS ONE policies on sharing data and materials.

## Conclusion

Nrf2 signaling promotes cancer stemness, migration, and expression of ABC transporter genes in sorafenib-resistant HCC cells.

## Introduction

Hepatocellular carcinoma (HCC) is the most common primary liver cancer and is a serious medical problem, as its incidence is continuing to grow. HCC development is a complex, multi-step process which ultimately results in inflammatory harm, hepatocyte necrosis/regeneration, and fibrotic deposits [1]. However, there are limitations to chemotherapy treatment of HCC. The drugs currently used for first-line systemic treatment, such as sorafenib and lenvatinib, can extend patient survival for only several months, mainly because of the development of resistance to these therapies [2].

Previous studies have reported potential mechanisms that lead to sorafenib-resistant HCC [3]. Nuclear receptor binding protein 2 (NRBP2) may increase HCC cell chemotherapy resistance by affecting expression of survival proteins in the Bcl2 and Akt pathways [4]. The histone demethylase lysine-specific demethylase 1 (KDM1A) can reduce therapeutic sensitivity in HCC by increasing the β-catenin pathway through activation of Wnt signaling [5]. Additionally, KRAS pathway accelerates RAF/ERK and PI3K/AKT signaling, which results in increased cell proliferation and suppressed apoptosis in sorafenib-resistant HCC cells [6].

Several studies have shown that cancer stem cells (CSCs) play a significant role in cancer recurrence and major resistance to molecularly targeted therapies. Recent work has indicated that HCC cells with stem cell-like characteristics, such as expressing the CSC surface markers CD44, EpCAM, CD133, and CD90, displayed resistance to sorafenib-induced cell death [7]. However, the mechanism related to acquisition of cancer stemness in sorafenib-resistant cells remains unclear [8].

Nuclear factor erythroid-derived 2-like 2 (Nrf2) signaling abnormalities are commonly found in a variety of cancers, including HCC, and are involved in tumorigenesis, tumor progression, and chemotherapy resistance [9]. Nrf2 helps to maintain the oxidative stress balance, and can promote the survival of cancer cells under xenobiotic toxicants by activating transcription of several anti-oxidative genes. The Keap1/Nrf2 pathway has been regarded as the main signaling cascade that regulates cellular defenses against oxidative stress. Furthermore, Nrf2 influences the tumor microenvironment by driving macrophage polarization into the M2 phenotype and promoting migration of cancer cells [10]. Under normal circumstances, Keap1 isolates and binds Nrf2 in the cytoplasm, leading to proteasome-mediated degradation of downstream genes [11]. In certain cases, Nrf2 is released from Keap1 and transferred to the nucleus, thereby activating ARE-mediated expression of detoxifying enzyme genes, including HO-1 [12]. HO-1 is involved in the regulation of NRF2-targeted ATP-binding cassette (ABC) efflux transporters (ABCC1, ABCG2, etc.) [13]. Additionally, Nrf2 induces the expression of glycolysis genes and takes part in the transcriptional regulation of genes important for stemness in cancer cells, which promotes malignancy [14]. The dark side of Nrf2 signaling was also described in cancer stem cells. Activated Nrf2 resulted in less ROS production and refractory response to drugs [15]. Nrf2, as a transcription factor, contributed to tumor generation of cancer stem cells by gene editing technics [16].

In this study, we investigated the mechanism responsible for sorafenib resistance in HCC cells, focusing on the Nrf2 signaling pathway. We examined if sorafenib-resistant HCC cells

displayed increased cell viability and invasive capacity after being stably cultured in sorafenib-supplemented medium. We tested the hypothesis that the induction of cancer stemness and the disruption of ABC efflux transporters related to Nrf2 signaling pathway was the mechanism.

## Materials and methods

### Cell culture

The human HCC cell line, Huh7, was purchased from the Riken Cell Bank (Tsukuba, Japan). DMEM (Life Technologies Japan Ltd., Tokyo, Japan) containing 10% FBS (Life Technologies Japan Ltd., Tokyo, Japan) was used to culture parental Huh7 cells. The sorafenib-resistant Huh7 cells were cultured in DMEM with 8.0μM sorafenib (sc-220125A, Santa Cruz, CA) dissolved in DMSO. The final concentration of DMSO was less than 0.1%.

### Establishment of sorafenib-resistant cell line

Sorafenib-resistant Huh7 cells were generated from the parental cells through continuous exposure to sorafenib-supplemented medium. Through increased selective pressure, the surviving Huh7 cells became resistant to sorafenib. These cells were transferred to medium that contained a higher concentration of sorafenib, which was increased by 0.5 μM weekly. Huh7 cells that acquired resistance were stably cultured in the medium with 8.0 μM.

### Cell viability assay

Cell viability was determined using cell-counting kit-8 (Dojindo Molecular Technologies, Inc. Tokyo, Japan) following the manufacturer's instructions. Cells were seeded at a density of $2 \times 10^4$ cells/well in a clear, flat-bottom 96-well plate (Falcon) andco-cultured with increasing dose of sorafenib for 48 hours, then incubated in 10% cell-counting kit-8 culture medium for 1–4 hours. Absorbance at 450 nm was then measured with a plate reader (SpectraMax i3, Molecular Devices, Tokyo, Japan). IC50 (concentration of sorafenib which is required for 50% inhibition) was calculated using GraphPad Prism 5 software (GraphPad Software, USA).

### Migration assay

A trans-well insert (Corning, NY. USA) with a pore size of 8 μm was used to conduct a migration assay. Cells ($0.5–1 \times 10^5$) were seeded in the upper chamber and allowed to attach for about 12 hours. These cells were gently washed three times with DMEM, and conditioned medium (DMEM + 5% FBS) was added. Conditioned medium was also added to the lower chamber. After 24 hours of incubation, any cells below the trans-well insert were fixed in 4% paraformaldehyde for 30 minutes, then stained in 0.2% crystal violet. Five random microscope fields of stained cells were counted.

### Wound healing assay

Parental and sorafenib-resistant Huh7 cells were plated in 6 well plates. Then confluent cells were scratched with a 10 μL pipette tip to create a ~0.5-mm-wide gap through the entirety of each cell layer. The medium was replaced with DMEM with or without 8.0μM sorafenib, where indicated. The scratch was observed using a microscope at 20× magnification for 48 hours. Wound healing was quantified using ImageJ (National Institutes of Health, USA).

## Colony formation assay

Two hundred parental and sorafenib-resistant Huh7 cells were seeded in 60mm cell culture dish (Thermo Scientific) and cultured in DMEM with or without 8.0 μM sorafenib for 3 weeks. Cells were then fixed in 4% paraformaldehyde and stained in 0.2% crystal violet. After washing, cell colony numbers were quantified.

## RNA isolation and real time PCR (RT-PCR)

Total RNA was isolated using an RNeasy Mini Kit (Qiagen, Hilden, Germany) following the manufacturer's instructions. The RNA concentration and ratio at A260/280 were detected by a NanoDrop spectrophotometer (Thermo Fisher Scientific). Complementary DNA (cDNA) was synthesized using a reverse transcription kit (Applied Biosystems, Foster City, CA, USA) with 2.5μg total RNA for every sample. RT-PCR was performed with 2μL/well in triplicate using a StepOnePlus Real-Time PCR System (Applied Biosystems). Human GAPDH (4352339E) was used as an internal control. The primer sequences are provided in S1 Table.

## SiRNA transfection

Sorafenib-resistant Huh7 cells were transfected with an Nrf2 siRNA (10 nmol/L, Applied Biosystems, S9492, Waltham, Massachusetts, USA), or a negative control siRNA (10 nmol/L, Applied Biosystems, Select Negative Control #1 siRNA, Waltham, MA, USA) using Invitrogen Lipofectamine RNAiMAX Transfection Reagent (Invitrogen, Thermo Fisher Scientific Inc., Waltham, MA, USA) following the manufacturer's instructions.

## Sphere forming assay

Huh7 cells cells were seeded by a density of 1000 cells/ml in a 24-well plate. Sphere culture medium was composed of serum free DMEM/F12 with 2% B27-supplement (20ng/ml human recombinant basic fibroblast growth factor (Peprotech, USA), 20ng/ml human recombinant epidermal growth factor (Peprotech, USA), 1% sodium pyruvate (WAKO, Japan), 1% MEM nonessential amino acids solution (Gibco, US), and 1% GlutaMax (Gibco, US) [17–19]. The spheres were observed using a microscope at 20× magnification after 7 days.

## Haematoxylin and eosin (H&E) staining

HCCs were cultured in chamber slides (Matsunami, Japan) for 3 days, then gently washed with PBS for 3 times fixed for 0.5h in 4% formaldehyde at the room temperature. For H&E staining, the nucleus were stained in blue by hematoxylin, and eosin stains the ECM and cytoplasm pink. Then the slides were observed with a microscope.

## Statistical analysis

Data analysis was performed using GraphPad Prism 5 software (GraphPad Software, USA) to conduct one-way ANOVA and Mann–Whitney U tests. All data are presented as the mean ± SD. A value of $p < 0.05$ indicates statistical significance.

# Results

## Sorafenib-resistant HCC cells acquired morphological and malignant changes

To establish a sorafenib-resistant Huh7 cell line, parental Huh7 cells were cultured in increasing doses of sorafenib and stably grown until reaching the target concentration of 8 μM. We

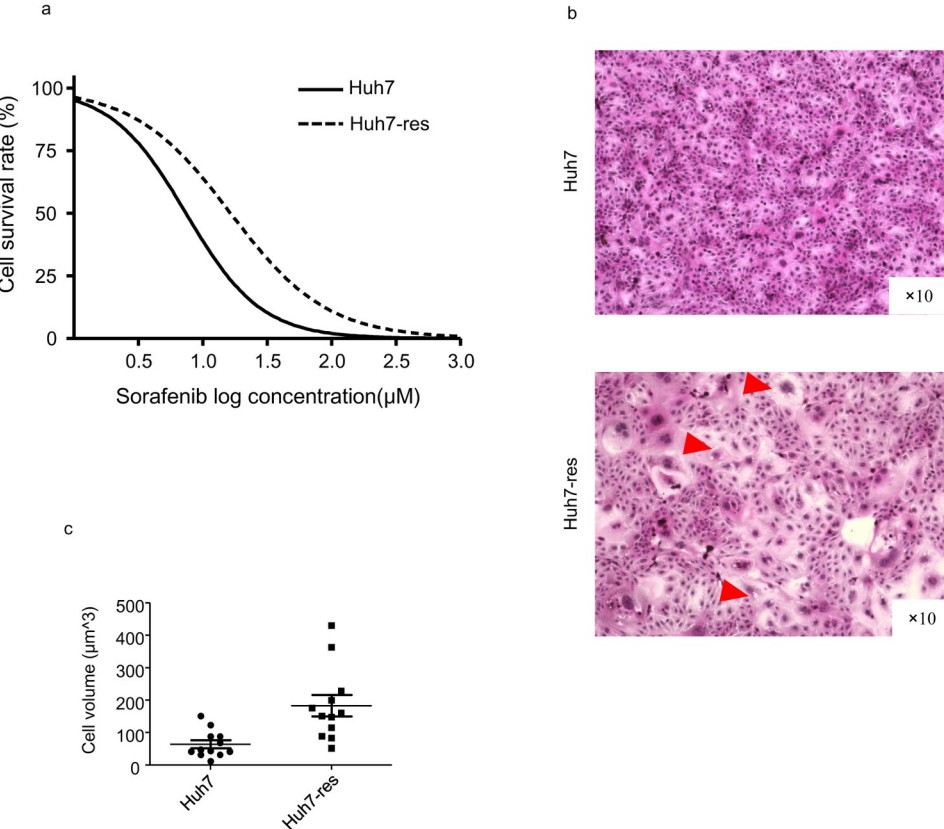

**Fig 1. Morphology of established sorafenib-resistant Huh7 cells.** a. Sorafenib-resistant Huh7 cells showed sorafenib IC50 increased compared with normal Huh7 cells. b. Morphology of sorafenib-resistant Huh7 cells compared with normal Huh7 cells. Compared with normal Huh7 cells, the drug-resistant cells are more heterogeneous, the cell volume increases, the cytoplasm is richer, and vacuole structures are indicated by (red Δ). c. Cell volumes were qualified with ImageJ (n = 12). P value<0.01.

then measured the chemotherapy IC50 of the cells and observed that the resistant cell line lost sensitivity to sorafenib treatment for 48 hours (Fig 1a). Results showed that IC50 of Huh7 was 10.68μM, IC50 of sorafenib resistant Huh7 was 16.48μM (P value<0.001). Compared with parental Huh7 cells, the sorafenib-resistant Huh7 cells displayed more heterogeneous morphology; the cell volume increased, the cytoplasm and vacuole structures were visible (Fig 1b and 1c).

We then examined malignant behavior in the cell lines. Migration and wound healing assays were performed to analyze the migration abilities of parental and sorafenib-resistant Huh7 cells in medium with or without sorafenib added. When cultured in regular DMEM, sorafenib-resistant Huh7 cells did not migrate at a statistically significantly different rate to the parental cells. However, in sorafenib-supplemented DMEM, the migration rate of parental Huh7 cells was significantly lower than that of sorafenib-resistant Huh7 cells (Fig 2a). A transwell assay was conducted to examine the invasion abilities of the cell lines. In both the DMEM and sorafenib-supplemented DMEM, the number of sorafenib-resistant Huh7 cells that moved into the lower chamber was significantly higher than that of parental cells, especially in sorafenib conditioned medium (Fig 2b). The expression of proliferation marker Ki67 was obviously increased in resistant cells (S1a Fig). The colony formation assay revealed that

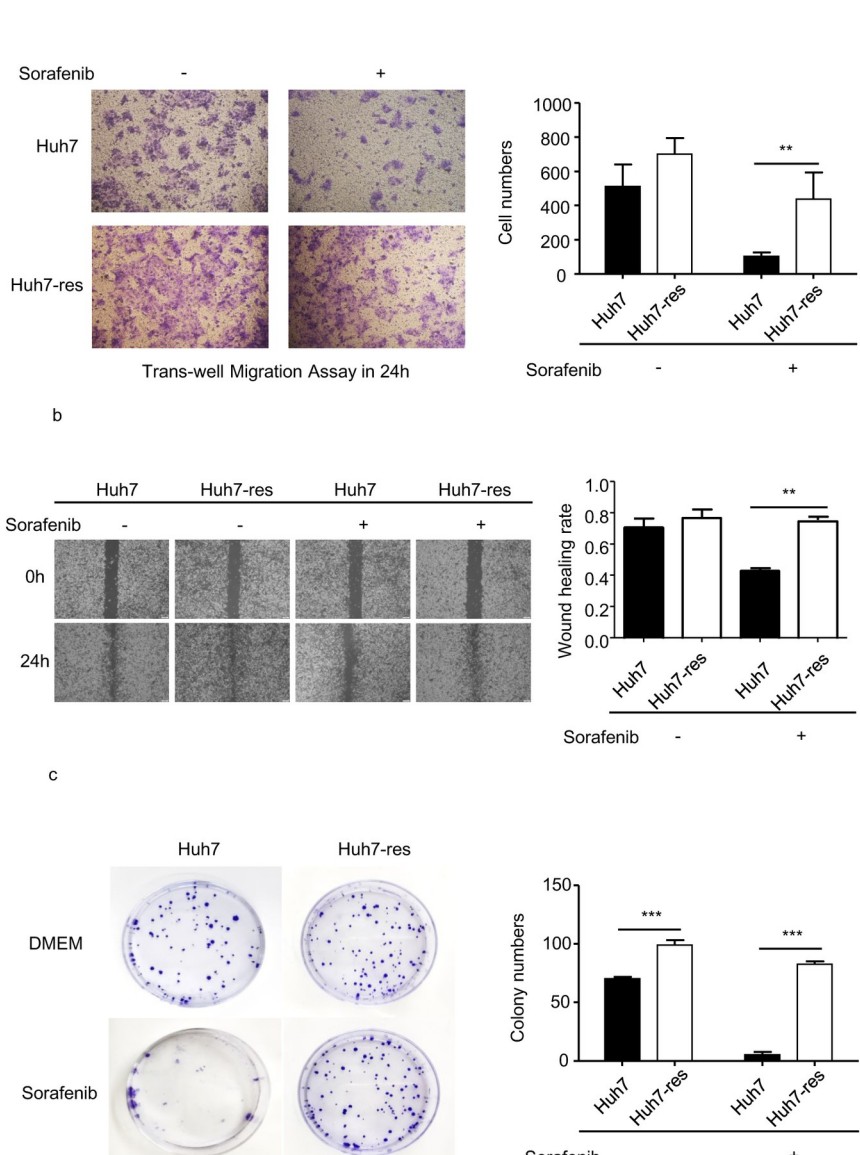

**Fig 2. Migration, invation and proliferation ability were detected in resistant Huh7 cells compared with Huh7 cells.** a. Invation ability was perfomed by tranawell assay. ImageJ quantification of cell numbers. (n = 4) * indicates P value<0.05) ** indicates P value<0.01,) *** indicates P value<0.001. b. Migration rate of resistant cell line compared with normal cells was perfomed by wound healing assay. ImageJ quantification of migration rate. (n = 3) * indicates P value<0.05) ** indicates P value<0.01,) *** indicates P value<0.001. c. Proliferation ability was perfomed by cloning forming assay. ImageJ quantification of group numbers. (n = 3) * indicates P value<0.05,) ** indicates P value<0.01,) *** indicates P value<0.001.

proliferation of sorafenib-resistant Huh7 cells, both with and without sorafenib in the medium, was also higher compared with parental cells (Fig 2c).

## Cancer stemness was enhanced in sorafenib-resistant HCC cells

Recently, it has been reported that CSCs are responsible for cancer recurrence and chemotherapy resistance [3]. To investigate this, we isolated total RNA from the parental and sorafenib-resistant Huh7 cells and used RT-PCR to examine expression levels of stemness-related markers,

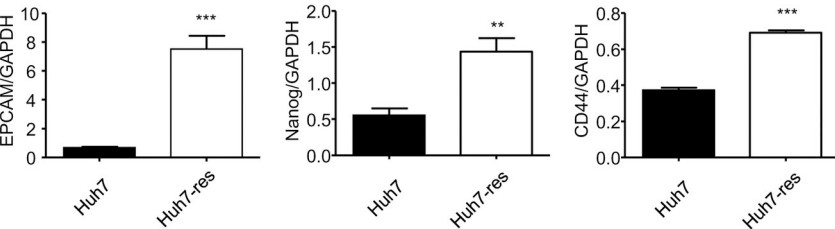

**Fig 3. Related mRNA expression of stemness markers were upregulated in sorafenib-resistant Huh7 cells.** PCR quantification of EpCAM, Nanog and CD44 mRNA expressed in sorafenib-resistant Huh7 cells compared with normal Huh7 cells, expressed for 48h cultured in DMEM (10%FBS). (n = 4) (* indicates P value<0.05 ** indicates P value<0.01 *** indicates P value<0.001).

including EPCAM, Nanog, and CD44. Sorafenib-resistant Huh7 cells exhibited increased expression of stemness-related genes compared with parental Huh7 cells (Fig 3), which reveals that stem-like properties were gained during the development of sorafenib resistance. In sphere forming assay, sorafenib resistance also mediated higher cancer stem like properties (S1b Fig). These results suggested that Huh7 cells gained stemness since resistance to sorafenib.

## The Nrf2 pathway was activated in sorafenib-resistant HCC cells

Previous work has demonstrated that the KEAP1/NRF2 pathway is one of the most frequently mutated pathways in human HCC and is relevant to chemotherapy drug resistance [13]. Because of this, we examined Nrf2 mRNA levels in parental and sorafenib-resistant Huh7 cells. Our results suggest that expression levels of Nrf2 and HO-1 in sorafenib-resistant Huh7 cells were significantly higher than levels in parental Huh7 cells (Fig 4a).

Other groups have shown that Nrf2 is released from Keap1 and then transferred to the nucleus, where it activates ARE-mediated expression of detoxifying enzyme genes, including HO-1. HO-1 is involved in the regulation of Nrf2-targeted ABC efflux transporters such as ABCC1 [13]. Expression levels of ABC transporters, including ABCA6, ABCB1, ABCC1, and ABCG2, were also significantly increased in sorafenib-resistant Huh7 cells (Fig 4b). The results showed that Nrf2 signaling was upregulated in sorafenib-resistant Huh7 cells.

## Knockdown of Nrf2 modulated expression of cancer stemness markers and ABC transporter genes in sorafenib-resistant HCC cells

To examine the relationship between the Nrf2 signaling pathway and sorafenib-resistance in HCC, Nrf2 expression was knocked down using a specific siRNA in sorafenib-resistant Huh7 cells, and a non-targeting siRNA was used as a control. Nrf2 knockdown reduced the expression of cancer stemness markers EPCAM, CD44 and Nanog in sorafenib-resistant Huh7 cells (Fig 5a). Expression levels of ABC transporters ABCB1, ABCC1, and ABCG2 were also reduced in Nrf2 siRNA-treated sorafenib-resistant Huh7 cells, but not ABCA6 (Fig 5b). The expression in protein level of Nrf2, HO-1, ABCC1 and ABCG2 showed the similar results (Fig 5c). Overall, these results suggest that the cells obtained resistance to sorafenib through the Nrf2 signaling pathway and that downregulating Nrf2 in the resistant cells reversed this progression.

## Knockdown of Nrf2 reduced malignant behavior in sorafenib-resistant HCC cells

To determine changes in cell viability in Nrf2 siRNA-treated sorafenib-resistant Huh7 cells, trans-well and wound healing assays were performed to identify the migration and invasion

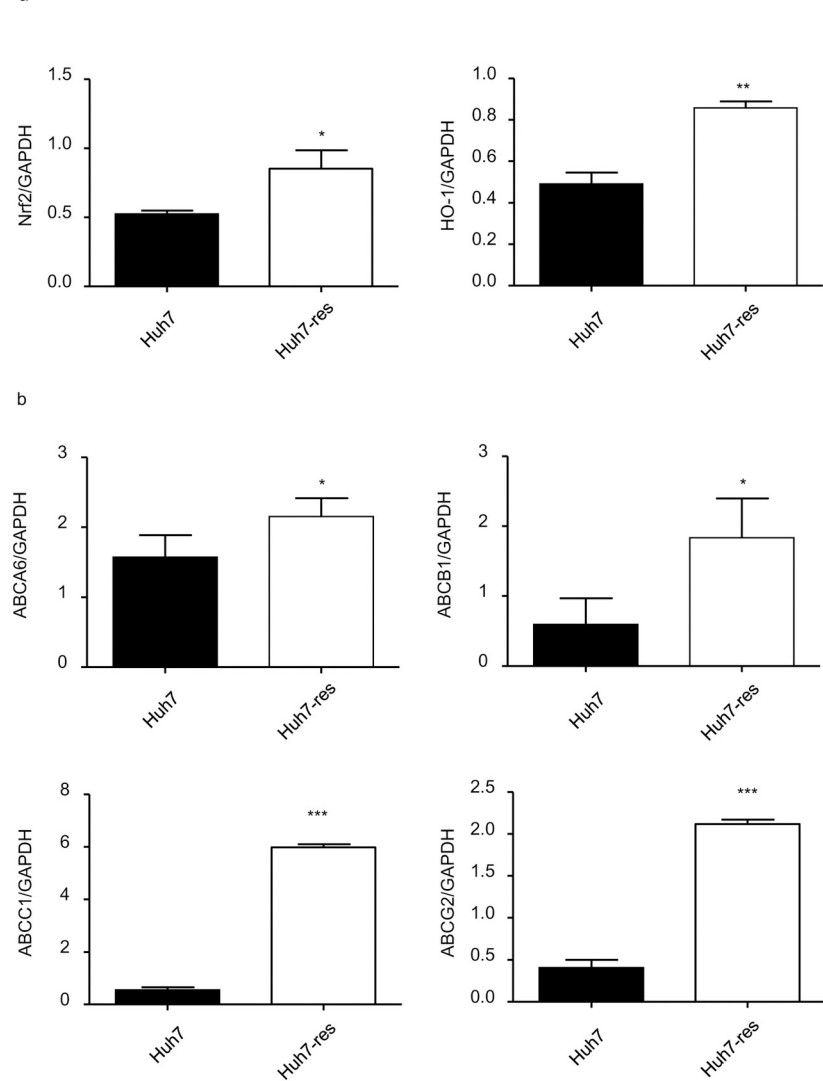

**Fig 4. Related mRNA expression of Nrf2 and ABC transporters were upregulated in sorafenib-resistant Huh7 cells.** a. PCR quantification of Nrf2 and HO-1 mRNA expressed in sorafenib-resistant Huh7 cells compared with normal Huh7 cells, expressed for 48h cultured in DMEM (10%FBS). (n = 4) * indicates P value<0.05) ** indicates P value<0.01) *** indicates P value<0.001. b. PCR quantification of ABCA6, ABCB1, ABCC1, ABCG2 mRNA expressed in Sorafenib-resistant Huh7 cells compared with normal Huh7 cells, expressed for 48h cultured in DMEM (10%FBS). (n = 4) * indicates P value<0.05) ** indicates P value<0.01) *** indicates P value<0.001.

ability of these cells in medium with or without sorafenib. The results suggested that the migration ability of sorafenib-resistant Huh7 cells was significantly decreased following Nrf2 knockdown (Fig 6a). Furthermore, Nrf2 knockdown reduced the invasion and proliferation ability of these cells (Fig 6b and 6c). These data revealed that targeting the Nrf2 signaling pathway may be a potential therapeutic strategy for the treatment of sorafenib-resistant HCC.

## Discussion

In this study, we established a sorafenib-resistant HCC cell line (Huh7) by increasing the dose of sorafenib in the medium and maintained stable growth until reaching the target concentration (8 uM). Sorafenib-resistant cells displayed more heterogeneity: increased cell volume,

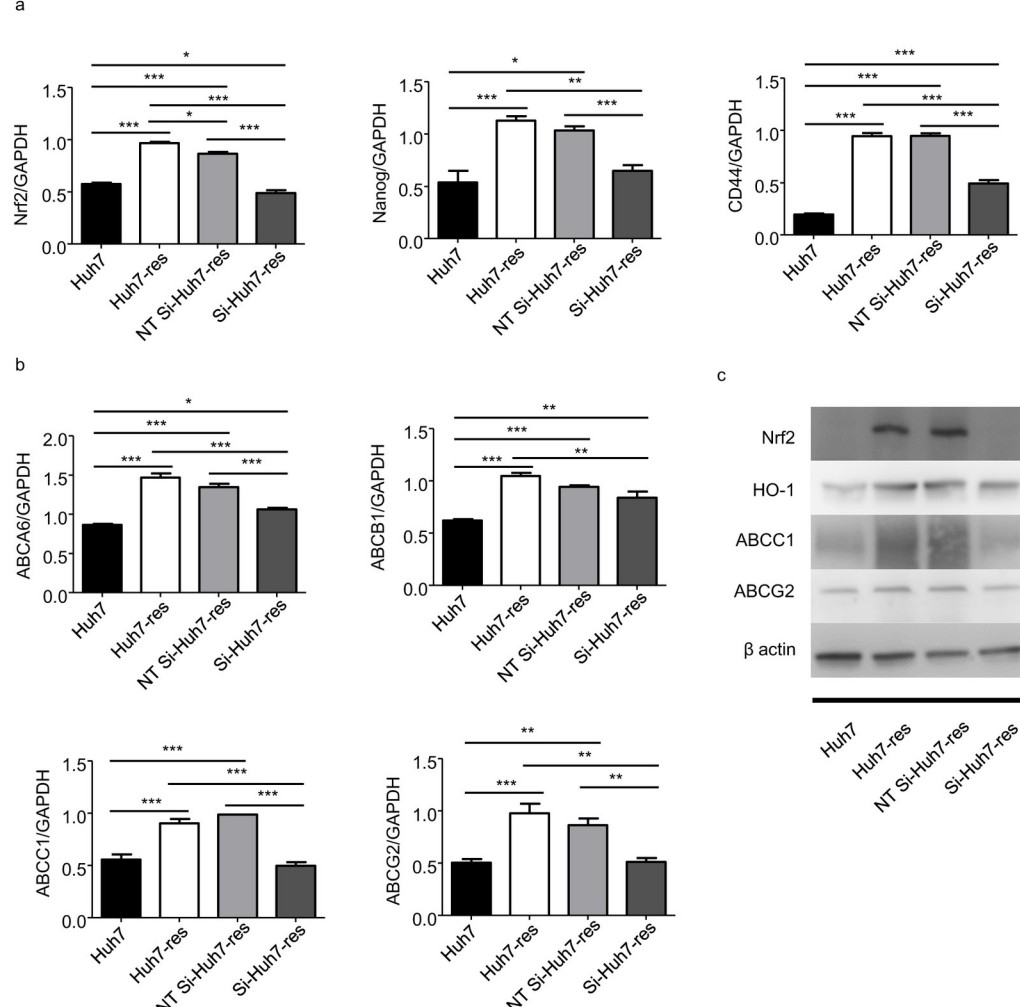

**Fig 5. Decreased expression of Nrf2 related genes and stemness markers in Nrf2 knocked down resistant Huh7 cells treated with siRNA.** a. PCR quantification of Nrf2, CD44, Nanog mRNA expressed in Nrf2 knocked down resistant Huh7 cells compared with sorafenib-resistant Huh7 cells, non-targeting control si-RNA treated resistant cells (NT Si-Huh7-res) and normal Huh7 cells, expressed for 48h cultured in DMEM (10%FBS). (n = 4) * indicates P value<0.05) ** indicates P value<0.01) *** indicates P value<0.001. b. PCR quantification of ABC tranporters ABCA6, ABCB1, ABCC1, ABCG2 mRNA expressed in Nrf2 knocked down resistant Huh7 cells compared with sorafenib-resistant Huh7 cells, NT Si-Huh7-res cells and normal Huh7 cells, expressed for 48h cultured in DMEM (10%FBS). (n = 4) * indicates P value<0.05) ** indicates P value<0.01) *** indicates P value<0.001. c. Levels of Nrf2 related protein (Nrf2, HO-1) and ABC transporters (ABCC1, ABCG2) in Nrf2 knocked down resistant Huh7 cells compared with sorafenib-resistant Huh7 cells, NT Si-Huh7-res cells and normal Huh7 cells were analyzed by western blotting.

richer cytoplasm, and the presence of vacuole structures. Migration, invasion, and mRNA expression of several CSC markers were significantly increased in the sorafenib-resistant Huh7 cells compared with the parental cell line. Malignancy and cancer stemness were enhanced in the sorafenib-resistant cells. Next, an siRNA was used to knock down Nrf2 expression in sorafenib-resistant cells. Consequently, the observed increases in proliferation, migration, invasion, and expression of stemness-related markers in sorafenib-resistant Huh7 cells were negated following Nrf2 knockdown. Furthermore, Nrf2 knockdown in these cells resulted in the downregulation of several ABC transporter genes.

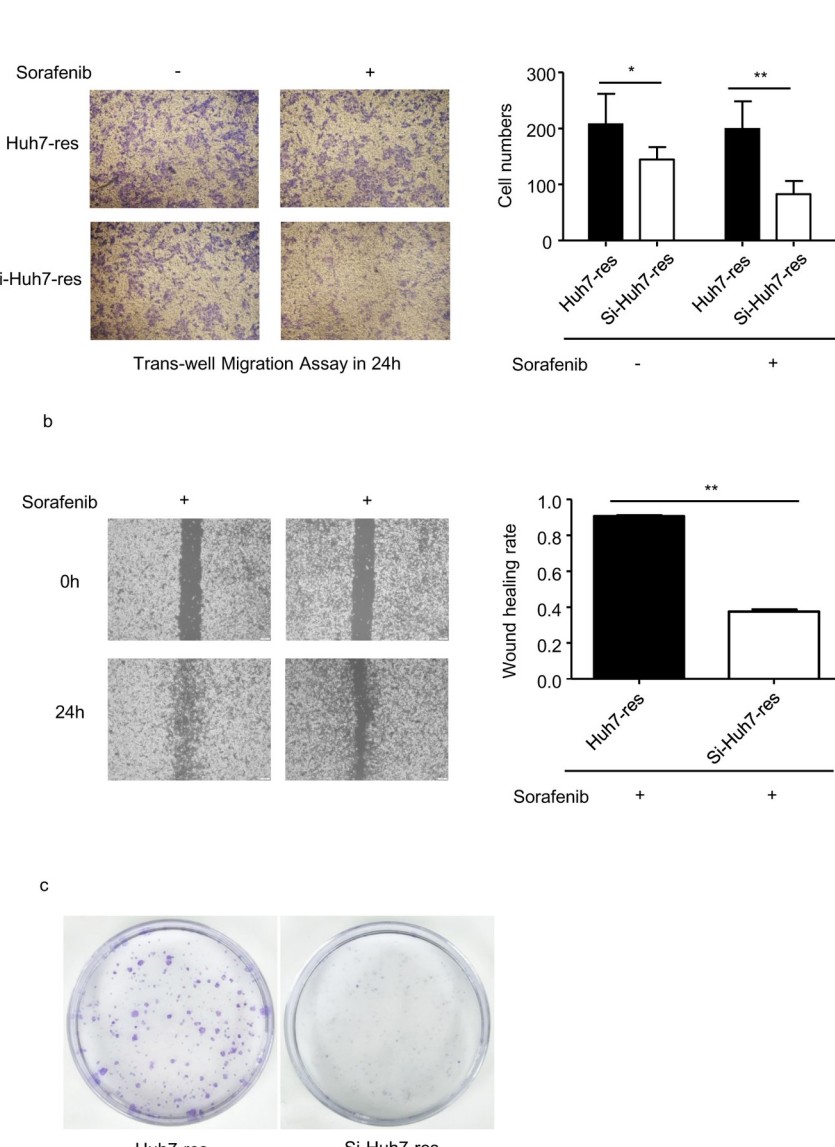

**Fig 6. Enhanced cell viability in resistant Huh7 cells were reversed in Nrf2 knocked down resistant Huh7 cells treated with siRNA.** a. Invation ability was perfomed by tranawell assay for sorafenib-resistant Huh7 cells and siRNA treated sorafenib-resistant Huh7 cells cultured in DMEM or sorafenib added DMEM. ImageJ quantification of cell numbers. (n = 4) * indicates P value<0.05) ** indicates P value<0.01,) *** indicates P value<0.001. b. Migration rate of sorafenib-resistant Huh7 cells and siRNA treated sorafenib-resistant Huh7 cells cultured in sorafenib added DMEM was perfomed by wound healing assay. ImageJ quantification of migration rate. (n = 3) * indicates P value<0.05) ** indicates P value<0.01,) *** indicates P value<0.001. c. Proliferation ability was perfomed in sorafenib-resistant Huh7 cells and Nrf2 knocked down sorafenib-resistant Huh7 cells by cloning forming assay.

Sorafenib blocks several tyrosine kinases, such as RAF, PDGFR, and VEGFR, by modulating their downstream signaling pathways and decreasing cell proliferation and survival rates [20]. In HCC patients with unresectable tumors, sorafenib treatment improved overall survival. However, the therapeutic effect of sorafenib is modest in HCC, only increasing median overall survival of patients by 3 months. Drug resistance is considered the main reason for the failure of sorafenib treatment in these patients [21]. Many studies have investigated the

mechanisms controlling sorafenib-resistance in cancer. NF-κB, Akt/ERK, Wnt/β-catenin, and mTOR signaling have all been implicated in sorafenib-resistance in several kinds of tumors [22–24].

As a common and redox-sensitive leucine zipper protein, Nrf2 can regulate cellular responses to oxidative and electrophilic damage [25]. Nrf2 is localized in the cytoplasm under steady-state conditions and is isolated by its inhibitor KEAP1 [14]. There is controversy regarding the role of Nrf2 in cancer, yet studies have shown that activating it can inhibit tumoregeneration [26]. However, other studies have described Nrf2 as being involved in chemotherapy resistance, which suggests that it acts in an oncogenic manner via regulation of downstream signaling molecules, like GLI1 and SUFU [14]. Additional work has shown that inhibiting Nrf2 with an siRNA reduced cancer cell resistance to gemcitabine, cisplatin, 5-FU, etoposide, camptothecin, and doxorubicin in several cancers by modulating ROS and the downstream efflux system [27]. The phosphoglycerate dehydrogenase (PHGDH) has been found as a critical driver for ROS activity, which is regulated by expression of Nrf2 via ATF4, and finally results in sorafenib resistance [28]. In the sorafenib-resistant HCC cell line we established, expression of Nrf2 and HO-1 were robustly increased compared with the parental cell line, which suggests an important role for the Nrf2 signaling pathway in sorafenib-resistant HCC.

Additionally, ABC transporters are involved in multi-drug resistance. ABC transporters are highly expressed in embryonic organisms and are important in chemical detoxification [29]. High activity of transporters can decrease the efficacious concentration of chemotherapeutic compounds in cancer cells. ABC transporters generally associated with chemoresistance are MDR1 (ABCB1), BRCP (ABCG2), and MRPs [30,31]. Several tyrosine kinase inhibitors, including sorafenib, can interact with ABC transporters such as ABCB1, ABCC1, ABCG2, and ABCC10. These ABC transporters behave as special pumps to influence intracellular drug concentration. These inhibitors are regarded as antagonists of these transporter proteins [32]. As ABC transporters are downstream of Nrf2, the absence of Nrf2 enhanced the sensitivity of osteosarcoma to doxorubicin, cisplatin, and sorafenib and resulted in decreased expression of ABCC3, ABCC4, and ABCG2 [13]. Nrf2 activates the cellular DNA damage response and regulates drug metabolism and resistance via downstream enzymes and transporters [33]. In our work, sorafenib-resistant HCC cells also expressed higher levels of ABC transporters, including ABCA6, ABCB1, ABCC1, and ABCG2. Additionally, Nrf2 knockdown in sorafenib-resistant Huh7 cells reduced the expression of ABCB1, ABCC1, and ABCG2.

Cancer stemness is involved in sorafenib-resistance in various cancers, including lung, brain, liver, and breast. Differentiation into heterogeneous cancer cells, self-renewal capacity, and chemotherapy resistance are all regarded as cancer stem-like properties [3]. In HBV associated HCC, HBV replicating cells displayed resistance to cisplatin and sorafenib via activated Wnt signaling through EpCAM, and exhibited increased expression of stemness-related markers (CD44, CD133, NANOG, OCT4, and SOX2) [21]. In our study, cancer stemness markers such as CD44, Nanog, EpCAM showed increased expression in sorafenib-resistant HCC cells. These results could not only explain the lack of sensitivity of CSCs to anti-cancer drugs, but also how the chemotherapy-resistant cells could gain stemness.

Nrf2 is highly expressed during CSC growth. In ovarian CSCs, knocking down Nrf2 inhibited stem-like properties such as highly expressed CSC markers [34]. Nrf2 can stimulate antioxidant response elements in CSCs, which helps to reduce ROS concentration, prolong cell survival, and support cancer cell chemoresistance [15]. In our work, Nrf2 and stemness markers were all highly expressed in sorafenib-resistant HCC cells. Conversely, knocking down Nrf2 led to decreased expression of stemness markers such as CD44 and Nanog. These data

suggested that the Nrf2 signaling pathway regulates stem-like markers in sorafenib-resistant HCC cells.

Taken together, our findings are the first to support the notion that the Nrf2 signaling pathway could be a therapeutic target in sorafenib-resistant HCC. However, there are some limitations since in vivo studies have not been conducted. To further validate our results, we plan to develop xenografts in mice using sorafenib-resistant HCC cells and then determine the treatment effect of those agents. Subcutaneous xenograft tumors will be analyzed for malignancy and changes in stemness-related genes.

In conclusion, sorafenib-resistant HCC cells obtained stemness characteristics and were more viable after being stably cultured in sorafenib-supplemented medium. These changes were induced by the Nrf2 signaling pathway. Targeting Nrf2 may be a potential therapeutic strategy for the treatment of sorafenib-resistant HCC.

## Supporting information

**S1 Fig. Proliferation and stem properties were detected in resistant cells.** a. PCR quantification of Ki67 mRNA expressed in Huh7 cells and sorafenib resistant Huh7 cells, expressed for 48h cultured in DMEM (10%FBS). (n = 4) * indicates P value<0.05 ** indicates P value<0.01) *** indicates P value<0.001. b. Sphere forming assay was performed in Huh7 cells and sorafenib resistant Huh7 cells, expressed for 7 days cultured in sphere culture medium.
(JPG)

**S2 Fig. Stemness related genes were detected in Huh7-res cells in the presence or absence of sorafenib.** PCR quantification of Nanog and CD44 mRNA expressed in Huh7-res cells, expressed for 48h cultured in DMEM (10%FBS) with the presence or absence of sorafenib. (n = 4) NS indicates P value≥0.05.
(JPG)

**S3 Fig.**
(TIF)

**S4 Fig.**
(TIF)

**S5 Fig.**
(TIF)

**S6 Fig.**
(JPG)

**S1 Table. The primer sequences used in this study.**
(DOCX)

## Author Contributions

**Conceptualization:** Mitsuo Shimada.

**Data curation:** Luping Gao, Shinichiro Yamada.

**Formal analysis:** Tetsuya Ikemoto.

**Methodology:** Kazunori Tokuda.

**Project administration:** Yuji Morine.

**Resources:** Katsuki Miyazaki.

**Software:** Yu Saito.

**Visualization:** Chie Takasu.

**Writing – original draft:** Luping Gao.

**Writing – review & editing:** Luping Gao.

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
