## [Decision Letter · Decision Letter 0]

8 Apr 2021

PONE-D-20-37563

Nrf2 signaling promotes cancer stemness, migration, and expression of ABC transporter genes in sorafenib-resistant hepatocellular carcinoma cells

PLOS ONE

Dear Dr. Morine,

Thank you for submitting your manuscript to PLOS ONE. After careful consideration, we feel that it has merit but does not fully meet PLOS ONE’s publication criteria as it currently stands. Therefore, we invite you to submit a revised version of the manuscript that addresses the points raised during the review process.

Please  respond to all reviewer comments.

We look forward to receiving your revised manuscript.

Kind regards,

Olorunseun Ogunwobi, MD, PhD

Academic Editor

PLOS ONE

Journal Requirements:

Thank you for stating in your Funding Statement:

This study was partly supported by Research Program on Hepatitis from the Japan Agency for Medical Research and Development (AMED) Grant Numbers JP19fk0210048 and JP20fk0210048. And co-authors (Shimada M) received a research grants from Grant-in-Aid for Scientific Research (B) (KAKENHI), Project code: 18K02871, during the conduct of the study.

Reviewers' comments:

Reviewer's Responses to Questions

**Comments to the Author**

1. Is the manuscript technically sound, and do the data support the conclusions?

Reviewer #1: Partly

Reviewer #2: Partly

2. Has the statistical analysis been performed appropriately and rigorously? 

Reviewer #1: Yes

Reviewer #2: I Don't Know

3. Have the authors made all data underlying the findings in their manuscript fully available?

Reviewer #1: Yes

Reviewer #2: Yes

4. Is the manuscript presented in an intelligible fashion and written in standard English?

Reviewer #1: Yes

Reviewer #2: Yes

5. Review Comments to the Author

Reviewer #1: The authors have shown that Nrf2 signaling promotes cancer stemness, migration, and expression of ABC transporter genes in sorafenib-resistant hepatocellular carcinoma cells. The work is interesting, however there are several drawbacks.

Major concern:

1. Introduction portion is not well written and needs to be improved.

2. You have given references related to KEAP1/ NRF2 pathway and the role of CSCs in cancer recurrence and chemotherapy resistance in the result section. These should be mentioned in the introduction section.

2. Figure quality is poor. I will suggest you to provide better figures for wound healing assays (Figure 2 and 6).

3. All the conclusions are based only on RT-PCR data. You should provide western blot data for further validation.

Minor mistake:

1. What do you mean by “ideal” confluency in the wound healing assay?

2. The sequence of siRNA and primers are missing.

3. Mention the amount of total RNA used in cDNA synthesis.

4. Please provide the PCR protocol in details.

Reviewer #2: In this manuscript, Gao et al. demonstrate that cancer stemness, migration/invasion and expression of ABC transporter genes are upregulated in the sorafenib-resistant hepatocellular carcinoma cells (HCC) they generated. Knockdown of Nrf2 reduced cancer stemness, migration/invasion and expression of ABC transporter genes in the sorafenib-resistant HCC cells. The authors propose that targeting Nrf2 signaling pathway has therapeutic potential in sorafenib-resistant HCC. This is an interesting work and it has been shown overexpression of Nrf2 in HCCs and Nrf2 regulates oxidative stress response and drives tumorigenesis. However, some additional experiments and modifications/data interpretation to the manuscript should be addressed to strengthen the discovery.

Fig.1A Is the difference of cell survival rate between Huh7 and Huh7-res cells statistically significant?

Fig.1B Methodological detail of Fig. 1B is missing. Furthermore, no precise measurement of the difference of cell volume or cytoplasm between Huh7 and Huh7-res cells and statistical analysis. In addition, please indicate vacuole structures.

Fig. 2A&B Fig2C showed higher proliferation of Huh7-res cells vs. Huh7 cell in the presence of sorafenib in a 3-week assay. Are there any proliferation data in short time period, with and without sorafenib? Does the proliferation difference affect the results of migration and wound healing assay in 24h?

Fig.3 What are the mRNA levels of EPCAM, Nanog, CD44 in tumorsphere assay, a three-dimensional culture setting between Huh-7 and Huh-res cells? In addition, this experiment lacks methodological detail for other researcher to repeat the experiment, what is the PCR primer set for EPCAM, Nanog and CD44. Anther question is what are the mRNA levels of EPCAM, Nanog, CD44 in the presence or absence of sorafenib in Huh7-res cells?

Fig. 4. What are the protein levels of Nrf2, HO-1, ABCC1 ABCG2 and ABCB1 and ABCA6? Western blot showing protein levels of these proteins will strengthen this part of the paper.

Fig. 5 A non-targeting siRNA control is not included, but that needs to demonstrate the reduction of Nrf2, CD44, Nanog was not because of transfection itself.

Fig.6 Similar to the point above, how cell proliferation was affected by knockdown Nrf2? Again, is the invasion or migration measured here because of cell proliferation?

Ultimately, all assays were performed in a single cell line, Huh7. Minimally these experiments should be repeated in at least one additional HCC cell line and its sorafenib-resistant cell line to establish how general this phenomenon is.

Overall, the publication of this manuscript at this time is a bit premature.

6. PLOS authors have the option to publish the peer review history of their article (what does this mean?). If published, this will include your full peer review and any attached files.

Reviewer #1: No

Reviewer #2: No

---

## [Author Response · Author response to Decision Letter 0]

29 May 2021

Thank you so much for your kind consideration of our manuscript. Also, I appreciate your and reviewers’ valuable comments to improve quality of our manuscript. According to the associate editor’s and reviewers’ comments, the manuscript has been fully revised.

Please consider this paper for publication in PLOS ONE. 

Again, your kind consideration on this matter would be greatly appreciated.

---

## [Decision Letter · Decision Letter 1]

2 Jul 2021

PONE-D-20-37563R1

Nrf2 signaling promotes cancer stemness, migration, and expression of ABC transporter genes in sorafenib-resistant hepatocellular carcinoma cells

PLOS ONE

Dear Dr. Morine,

Thank you for submitting your manuscript to PLOS ONE. After careful consideration, we feel that it has merit but does not fully meet PLOS ONE’s publication criteria as it currently stands. Therefore, we invite you to submit a revised version of the manuscript that addresses the points raised during the review process.

Please ensure that all comments and concerns included in the reports from both reviewers are fully addressed.

We look forward to receiving your revised manuscript.

Kind regards,

Olorunseun Ogunwobi, MD, PhD

Academic Editor

PLOS ONE

Journal Requirements:

Reviewers' comments:

Reviewer's Responses to Questions

**Comments to the Author**

1. If the authors have adequately addressed your comments raised in a previous round of review and you feel that this manuscript is now acceptable for publication, you may indicate that here to bypass the “Comments to the Author” section, enter your conflict of interest statement in the “Confidential to Editor” section, and submit your "Accept" recommendation.

Reviewer #1: All comments have been addressed

Reviewer #2: All comments have been addressed

2. Is the manuscript technically sound, and do the data support the conclusions?

Reviewer #1: Partly

Reviewer #2: Yes

3. Has the statistical analysis been performed appropriately and rigorously? 

Reviewer #1: Yes

Reviewer #2: Yes

4. Have the authors made all data underlying the findings in their manuscript fully available?

Reviewer #1: Yes

Reviewer #2: Yes

5. Is the manuscript presented in an intelligible fashion and written in standard English?

Reviewer #1: Yes

Reviewer #2: Yes

6. Review Comments to the Author

Reviewer #1: The manuscript has been improved significantly. However, you have to take care of a few things to improve it further.

1. The figures of wound healing/ migration assays are still not clear.

2. The HO-1 western blot image is not good.

3. The fonts used in the figures are hazy. Please try to make it clearly visible. The supplementary figures' fonts are better than the fonts used in the main figures of the manuscript.

4. Please use the same font type throughout your manuscript.

5. Some typos are there in the manuscript, for eg,

page 58, line 3: “Nrf2 helps maintain the oxidative stress balance”

page 63: “asaay”

page 64: “Nuceis”

Reviewer #2: The authors have clearly responded to my questions and made the necessary changes. I would like to recommend the revised version of the manuscript for publication. Minor points/comments regarding the figures:

Fig. 1a In the text it states: The IC50 of Huh7 was 10.68μM, IC50 of sorafenib resistant Huh7 was 16.48μM; however, in the figure the x axis label for Sorafenib log concentration is nM.

Fig. 2a right panel missing "-" in Sorafenib treatment line.

Fig. 6b: the quality of the figure needs improvement.

7. PLOS authors have the option to publish the peer review history of their article (what does this mean?). If published, this will include your full peer review and any attached files.

Reviewer #1: No

Reviewer #2: No

---

## [Author Response · Author response to Decision Letter 1]

21 Jul 2021

First of all, thank you so much for your kind consideration of our manuscript. Also, I appreciate your and reviewers’ valuable comments to improve quality of our manuscript. According to the associate editor’s and reviewers’ comments, the manuscript has been fully revised.

---

## [Decision Letter · Decision Letter 2]

16 Aug 2021

Nrf2 signaling promotes cancer stemness, migration, and expression of ABC transporter genes in sorafenib-resistant hepatocellular carcinoma cells

PONE-D-20-37563R2

Dear Dr. Morine,

We’re pleased to inform you that your manuscript has been judged scientifically suitable for publication and will be formally accepted for publication once it meets all outstanding technical requirements.

Kind regards,

Olorunseun Ogunwobi, MD, PhD

Academic Editor

PLOS ONE

Additional Editor Comments (optional):

Reviewers' comments:

Reviewer's Responses to Questions

**Comments to the Author**

1. If the authors have adequately addressed your comments raised in a previous round of review and you feel that this manuscript is now acceptable for publication, you may indicate that here to bypass the “Comments to the Author” section, enter your conflict of interest statement in the “Confidential to Editor” section, and submit your "Accept" recommendation.

Reviewer #1: All comments have been addressed

Reviewer #2: All comments have been addressed

2. Is the manuscript technically sound, and do the data support the conclusions?

Reviewer #1: Yes

Reviewer #2: Yes

3. Has the statistical analysis been performed appropriately and rigorously? 

Reviewer #1: Yes

Reviewer #2: Yes

4. Have the authors made all data underlying the findings in their manuscript fully available?

Reviewer #1: Yes

Reviewer #2: Yes

5. Is the manuscript presented in an intelligible fashion and written in standard English?

Reviewer #1: Yes

Reviewer #2: Yes

6. Review Comments to the Author

Reviewer #1: The authors have addressed al the comments and the manuscript has been improved significantly. I will suggest you to provide the sequences of siRNAs used in this study.

Reviewer #2: The authors have made the changes that I've recommended. The revised version of the manuscript is suitable for publication.

7. PLOS authors have the option to publish the peer review history of their article (what does this mean?). If published, this will include your full peer review and any attached files.

Reviewer #1: No

Reviewer #2: No

---

## [Editor Report · Acceptance letter]

25 Aug 2021

PONE-D-20-37563R2 

Nrf2 signaling promotes cancer stemness, migration, and expression of ABC transporter genes in sorafenib-resistant hepatocellular carcinoma cells 

Dear Dr. Morine:

I'm pleased to inform you that your manuscript has been deemed suitable for publication in PLOS ONE. Congratulations! Your manuscript is now with our production department. 

Kind regards, 

on behalf of

Dr Olorunseun Ogunwobi 

Academic Editor

PLOS ONE